# Identification of pattern mining algorithm for rugby league players positional groups separation based on movement patterns

Victor Elijah Adeyemo [1,2,3,4] *, Anna Palczewska[1], Ben Jones[2,3,4,5,6], Dan Weaving[2]

**1** School of Built Environment, Engineering and Computing, Leeds Beckett University, Leeds, United Kingdom, **2** Carnegie School of Sport, Leeds Beckett University, Leeds, United Kingdom, **3** England Performance Unit, Rugby Football League, Manchester, United Kingdom, **4** Leeds Rhinos Rugby League Club, Leeds, United Kingdom, **5** School of Behavioural and Health Science, Faculty of Health Sciences, Australian Catholic University, Brisbane, QLD, Australia, **6** Division of Physiological Sciences and Health through Physical Activity, Lifestyle and Sport Research Centre, Department of Human Biology, Faculty of Health Sciences, University of Cape Town, Cape Town, South Africa

* v.adeyemo@leedsbeckett.ac.uk

**Data Availability Statement:** The GPS data were obtained (from clubs through the Rugby Football League) and are not available publicly. However,

## Abstract

The application of pattern mining algorithms to extract movement patterns from sports big data can improve training specificity by facilitating a more granular evaluation of movement. Since movement patterns can only occur as consecutive, non-consecutive, or non-sequential, this study aimed to identify the best set of movement patterns for player movement profiling in professional rugby league and quantify the similarity among distinct movement patterns. Three pattern mining algorithms (*l*-length Closed Contiguous [LCCspm], Longest Common Subsequence [LCS] and AprioriClose) were used to extract patterns to profile elite rugby football league hookers (n = 22 players) and wingers (n = 28 players) match-games movements across 319 matches. Jaccard similarity score was used to quantify the similarity between algorithms' movement patterns and machine learning classification modelling identified the best algorithm's movement patterns to separate playing positions. LCCspm and LCS movement patterns shared a 0.19 Jaccard similarity score. AprioriClose movement patterns shared no significant Jaccard similarity with LCCspm (0.008) and LCS (0.009) patterns. The closed contiguous movement patterns profiled by LCCspm best-separated players into playing positions. Multi-layered Perceptron classification algorithm achieved the highest accuracy of 91.02% and precision, recall and F1 scores of 0.91 respectively. Therefore, we recommend the extraction of closed contiguous (consecutive) over non-consecutive and non-sequential movement patterns for separating groups of players.

## Introduction

Big data in sports are often gathered through wearable sensors such as Global Positioning Systems [GPS] [1] and video-sourced match events extracted by an expert analyst(s) [2]. Importantly, wearable and video-based forms of sport-related big data usually exist as a collection of

the anonymised data used for classification modelling is privately available upon request.

**Funding:** The authors received no specific funding for this work.

**Competing interests:** The authors have declared that no competing interests exist.

ordered sequences of match activities [3, 4]. These provide information regarding "when", "who", "what", and "where" activities occurred [3]. Wearable sensors are worn by players to collect on-field activities (e.g., positional data, speeds) during training and or competition. Data collected via wearable sensors or video-sourced match events facilitate the creation of performance indicators for understanding physical, technical and tactical demands on players [5]. Examples of the use of performance indicators include preparing athletes for transition between levels [6], identification of skills for talent development [7], injury prevention and recovery [8], opposition analysis [3] and classification of players into competition levels based on playing positions [9] and they have been widely used to derive actionable insights for making data-driven decisions. However, the insights provided by these performance indicators are currently based on aggregated physical, technical and tactical demands either across a whole match or for specific periods within the game. This can be inadequate because each performance indicator typically accounts for a single activity or event (e.g., pass, shots, total distance covered, average match speed) without providing the context and or explanation of how such activity was performed.

Frequent pattern mining algorithms have been applied in sports in various contexts such as automatic tactics detection in soccer matches [10], athlete performance monitoring [11] and discrimination of non-scoring and scoring outcomes between attacking and defending rugby union teams [12]. Nowadays, sequential pattern mining algorithms [13, 14] are applied to sports data to profile players' movements. Player movement profiling has become an interesting research area because it offers an alternative view to understanding match demands by concurrent evaluation of the speeds, changes in speeds and turning angles completed by players at any point in time. It helps to identify frequent groups of movements performed by players and uncover how often those groups of movements were performed. Sweeting et. al. [13] proposed the first framework for player movement profiling. The authors profiled players' movement by finding movement sequences that occurred frequently from elite international-level female netball players' Radio Frequency (RF) data during four competitive matches. The method is based on grouping similar movement strings into 25 clusters using the hierarchical clustering technique and applying the longest common subsequence (LCS) [14] algorithm to extract the longest common movement patterns from each cluster.

White et. al. [15] suggested that the Sweeting et. al. [13] framework is not stable because it produces different movement patterns for the same set of movement strings in consecutive runs. Thus, they addressed it by developing the Sequential Movement Pattern-mining (SMP) framework and ran stability tests of both frameworks on the same set of rugby league elite players' movement strings. The SMP framework was more stable between the two existing and evaluated frameworks for profiling athletes' movement patterns. The SMP framework was applied by Collins et. al. [16] to quantify movement patterns and identify the differences among three rugby league competitions (i.e., International Rugby League, Super League (semi-)Finals and Super League regular season). The study [16] reported that no movement pattern was unique to a single competition level. The analysis of decomposed extracted movement patterns (i.e., movement units) using linear discriminant analysis (LDA) revealed low velocities with mixed turning angles and acceleration characterized the movement units that most differentiate the competitions. Despite the robustness and stability of the SMP framework for discovering movement sequences, the total number of obtainable extracted patterns is limited to the number of identified clusters. More importantly, only the longest common pattern per cluster is outputted while other interesting patterns are discarded. This may have influenced the profiled movement patterns and units that differentiate the competition levels in Collins et. al. [16] study.

The study of Adeyemo et. al. [17] addressed the above-mentioned limitations of the SMP framework [15] by proposing and developing a new algorithm called *l*-length closed contiguous sequential pattern mining algorithm (i.e. LCCspm). We developed the algorithm because the existing Closed Contiguous Sequential Pattern mining (CCSpan) algorithm [18] may not produce usable movement patterns for sporting contexts and could not scale well on large sets of (lengthy) players' movement sequences. The study [17] used LCCspm to find frequently occurring (with specified maximal length) closed contiguous movement patterns from sets of movement sequences of five England Rugby Football League (RFL) Super League team matches and closed contiguous match-event patterns from a set of match-event sequences of soccer national teams' players that participated in men's FIFA 2018 World cup. The experimental results demonstrated that LCCspm scaled better, ran faster and use lower memory than the Closed Contiguous Sequential Pattern mining (CCSpan) algorithm for mining closed contiguous patterns [17]. More so, LCCspm was able to identify a large number of frequent movement patterns based on a user-defined length of patterns and user-specified support threshold.

The importance of extracting movement patterns [15, 16] from discretized time-series physical data is that it helps to find and reveal groups of movement activities performed by players, unlike the physical, technical, tactical performance indicators that only account for the accumulation of single activities. The extraction of movement patterns to quantify players' completed movement activities provide granular information about match-based activities with more ease in comparison with the laborious activities involved in expertly coded video analysis. Extracted movement patterns provide the context lacking in accumulated activities reported by physical, technical and tactical indicators [17]. More so, the extracted movement patterns can enhance the specificity of training programmes [16]. However, the investigation of which type of movement pattern (mining algorithm) is best for rugby league players' movement profiling is yet to be explored.

Since pattern mining algorithms available for player movement profiling can either extract sequential (i.e., consecutive and non-consecutive) or non-sequential movement activities (i.e., movement patterns), this study is motivated to investigate which pattern mining algorithm provides the best type of movement pattern to profile players' movements in the context of player positions and investigate the similarity of these patterns. For instance, the LCS algorithm of the SMP framework [13, 15] identifies the longest common movement patterns with omissions of performed activities while still retaining the sequential order of movement occurrences and allows repetition of movement activities within a pattern. On the other hand, the LCCspm algorithm [17] identifies the user-defined lengths of frequent and closed contiguous movement patterns where the movement patterns are strictly adjacent and without omission of any movement activities. Another pattern mining algorithm is the AprioriClose algorithm [19] that can discover frequent and closed patterns as movement patterns. Frequent closed movement patterns identify movement patterns with omissions of performed activities, do not allow repetition of movement activities within a pattern and the performed activities are not in sequential order but in lexicographical order.

The context of separating players into playing positions was considered because it is reported to help with talent identification and recruitment [20], customized training [21] as well as players' performance profiling [22] among others. Two rugby league playing positions (i.e. hookers and wingers) were selected based on their known differences in tactical roles during matches and study [23] also revealed that hookers and wingers share nearly similar average body weight but perform distinct tactical roles and different on-field activities (e.g., 15-m and 40-m sprints). The separation of rugby league players into these two playing positions (i.e., hookers and wingers) based on various types of profiled movement patterns will identify the best type of movement patterns for profiling rugby league players into positions. Additionally,

it will assist in the identification of specific movement patterns performed by players within each playing position and how often those movement patterns were performed.

Therefore, this study aimed to identify which pattern mining algorithm extracts the best type of movement patterns to profile players into two rugby league playing positions (i.e., hookers and wingers). To achieve this aim, (1) the set of unique movement patterns per algorithm were quantified for similarity; (2). overlapping movement patterns among the pattern mining algorithms (and playing positions) were investigated; and (3) classification modelling and evaluation was conducted to measure the extent of separation each type of movement pattern can provide and thus identify the best pattern mining algorithm for profiling rugby league players into playing positions.

## Method

### Overview

An observational repeated measures design was used in which 10Hz GPS data from 50 elite male Rugby Football League players consisting of 12 teams that participated in 319 fixtures within the 2019 and 2020 seasons were collected via wearable sensors (Catapult S5, Catapult Innovations, Melbourne, Australia) worn during matches. Two playing positions were selected hookers (n = 22) and wingers (n = 28). A total of 1,036 total observations (hookers = 500 and wingers = 536) were included which represent players' movement sequence per fixture. The three types of obtainable movement patterns were extracted from processed GPS data by applying three pattern mining algorithms. Five machine learning classification algorithms were implemented and evaluated for each set of pattern, the evaluation results were used to decide which set gives the best separation for the playing positions. Also, movement patterns extracted by each algorithm were analysed for similarity and overlaps. The experimental framework depicted in Fig 1 illustrates an overview of data collection, processing and analysis methods. This study received the approval of the University Ethics Committee and obtained written informed consent from the organisation representing all participants. The GPS data were obtained (from clubs through the Rugby Football League) and are not available publicly.

### Data and processing

The method for generating movement sequences from global positioning systems data as published by [15] was followed to obtain sets of movement sequences. An example of GPS data and the result of the discretization is depicted in Table 1. Micro-sensor units including global positioning systems sampling at 10Hz [24] captured 50 elite male Rugby Football League players' physical demands (i.e., acceleration and velocity) and tracking data (i.e., latitude and longitude). GPS data of players with no recorded velocity, acceleration, tracking values were excluded. The velocity, acceleration and turning angle were extracted from GPS data and were further discretized using thresholds presented in Table 2 as published by [15].

Concatenation of velocity, acceleration and turning angle descriptors created the *movement unit* every 0.1s (10Hz) (Table 1, column "MovementUnit") which were assigned to a *movement unit character* (Table 1, column "A"). An example of movement sequence from Table 1 is the sequential concatenation of the movement unit characters in column "A" i.e., "ijfeikhddb". Inactive periods as proposed by [15] were filtered out from each player's continuous movement sequence resulting in obtaining a set of discrete movement sequences for each player per match. A total of 1,036 sets of discrete movement sequences were created representing all players' movement sequences per fixture (i.e., player-per-fixture granularity). Movement patterns were extracted from each set of discrete movement sequences and represented the frequent recurring movement patterns for each player within a match.

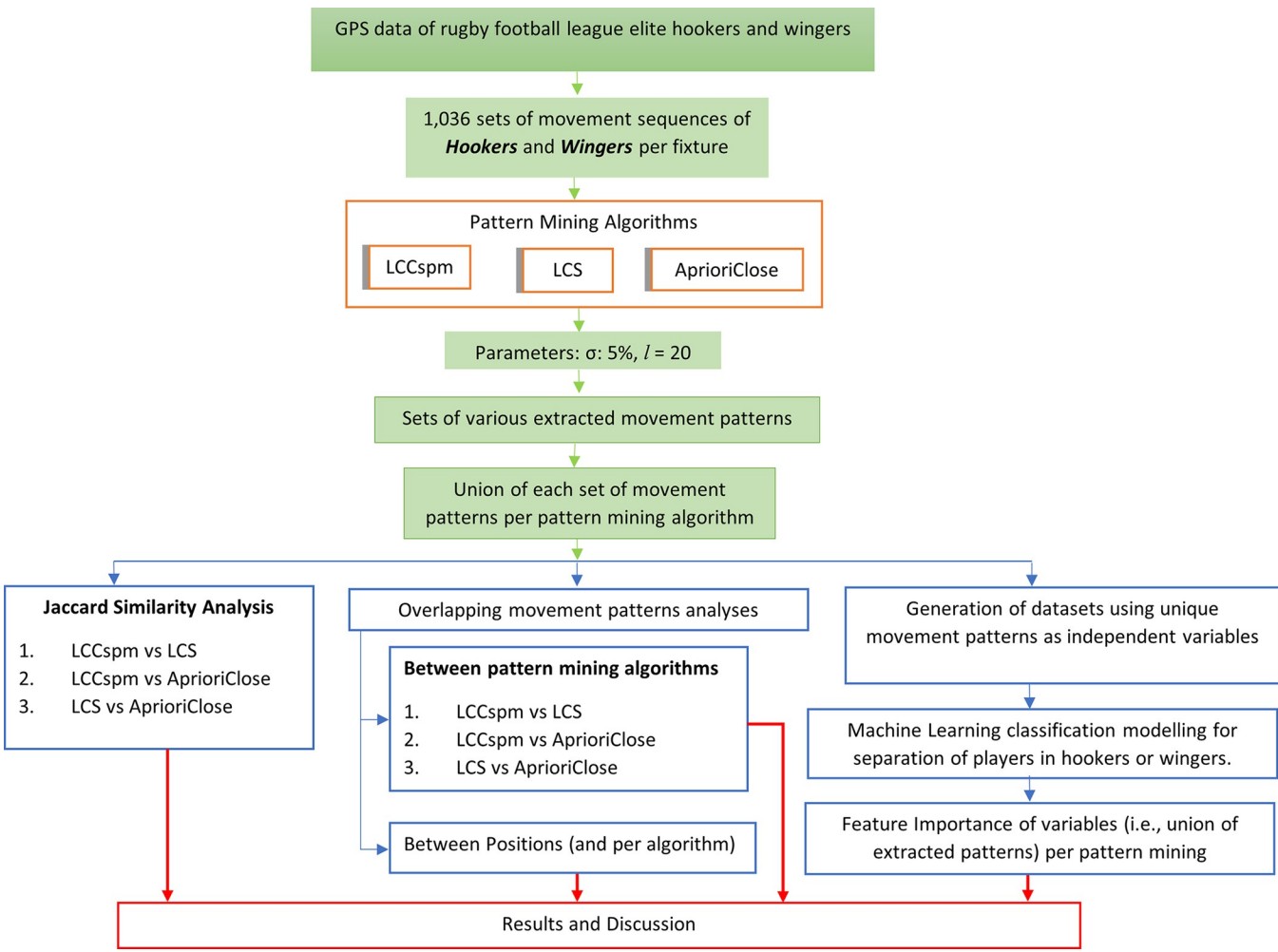

**Fig 1. Experimental framework.**

**Table 1. Example of processing GPS data into movement sequence.**

| Seconds | Vel Descr | Accel Descr | T.A. Descr | Movement Unit | A |
|---|---|---|---|---|---|
| 1469 | Walk | Acceleration | Straight | WalkAccelerationStraight | i |
| 1469.1 | Walk | Acceleration | Acute-Change | WalkAccelerationAcute-Change | j |
| 1469.2 | Walk | Neutral | Acute-Change | WalkNeutralAcute-Change | f |
| 1469.3 | Walk | Neutral | Straight | WalkNeutralStraight | e |
| 1469.4 | Walk | Acceleration | Straight | WalkAccelerationStraight | i |
| 1469.5 | Walk | Acceleration | Large-Change | WalkAccelerationLarge-Change | k |
| 1469.6 | Walk | Neutral | Backwards | WalkNeutralBackwards | h |
| 1469.7 | Walk | Deceleration | Backwards | WalkDecelerationBackwards | d |
| 1469.8 | Walk | Deceleration | Backwards | WalkDecelerationBackwards | d |
| 1469.9 | Walk | Deceleration | Acute-Change | WalkDecelerationAcute-Change | b |

**Table 2. The movement descriptors and threshold assignment values.**

| Velocity Descriptor | Velocity Threshold (m.s$^{-1}$) | Acceleration Descriptor | Acceleration Threshold (m.s$^{-2}$) | Turning Angle Descriptor | Turning Angle Threshold ($\Theta$) |
|---|---|---|---|---|---|
| Walk | 0.00 to <1.70 | Deceleration | Min accel to $\leq$ -0.20 | Straight | 0.00 to <10.00 |
| Jog | $\geq$ 1.70 to $\leq$ 3.90 | Neutral | >-0.20 to <0.20 | Acute-change | $\geq$ 10.00 to <45.00 |
| Run | >3.90 to <5.00 | Acceleration | $\geq$ 0.20 to max accel | Large-change | $\geq$ 45.00 to <90.00 |
| Sprint | $\geq$ 5.00 | N/A | N/A | Backwards | $\geq$ 90.00 to 180.00 |

## Pattern mining algorithms

To extract frequent movement patterns for the considered playing positions, three pattern mining algorithms were used: (1) LCS algorithm: existing and used within SMP framework [15], (2) LCCspm: a new algorithm, outperforming other closed-contiguous pattern mining algorithms [17] and (3) AprioriClose: a frequent closed pattern mining algorithm [19]. The LCS algorithm of the "SMP" has no parameter. The LCCspm algorithm [17] has two parameters: support (determines patterns' frequency) and length (determines patterns' maximum length). The AprioriClose [19] has a support parameter.

The support parameter of both the LCCspm and Apriori close algorithms was set to 5 percent to extract a large number of frequent movement patterns, because a high support threshold will identify few frequent patterns [25], from the player's discrete movement sequences (i.e., active periods within a match). LCCspm length parameter was set to 20 (i.e. 2 seconds time-frame) to ensure more and longer patterns are extracted. Meanwhile, the movement patterns extracted by AprioriClose and LCS algorithms were later filtered to exclude patterns containing more than 20 items. The studies [12, 17] used similar parameter values to enable the extraction of large and longer-length frequent patterns from rugby union and rugby league data.

Following the extraction of sets of user-defined length and frequent movement patterns from each set of 1,036 discrete movement sequences, a set of unique movement patterns was derived by computing the mathematical union of all sets of extracted movement patterns, per pattern mining algorithm. The sets of unique movement patterns (per algorithm) were subjected to further analysis discussed in the section below.

## Selection analysis of (movement) pattern mining algorithms

Three steps were taken to identify the best pattern mining algorithm to extract movement patterns for profiling rugby league players into playing positions. First, similarity analysis of the different sets of unique movement patterns obtained using each pattern mining algorithm was considered. The analysis of overlap movement patterns among pattern mining algorithms as well as within each playing position was also carried out. Lastly, the separation of players into playing positions (hookers and wingers) based on the different sets of extracted movement patterns was conducted and measured.

**Jaccard analysis.** Jaccard similarity measure [26] enables exact matching of patterns between two sets and was used to quantify the similarity among the groups of extracted patterns. It is computed as:

$$J(X, Y) = |X \cap Y| / |X \cup Y| \tag{1}$$

Jaccard similarity measure values ranged from 0 to 1, where 0 indicates no similarity and 1 indicates an exact match. The similarity among the sets of unique movement patterns (i.e., the union of all extracted movement patterns for all player-match levels) identified by each pattern mining algorithm was quantified by the Jaccard similarity measure.

**Overlap movement patterns.** Overlap movement patterns between two sets of movement patterns were identified using the exact matching method. Overlapping unique movement patterns between pairs of pattern mining algorithms were identified. Each pair's top and bottom patterns were checked for overlap by comparing the most frequent-50 and least frequent-50 patterns from each algorithm, representing one-third of the lowest set of extracted movement patterns, and then visualize. This was carried out to identify where the overlapping movement patterns are located. A further analysis was carried out on the identified overlapped movement patterns to discover those patterns performed by players of each playing position. Also, the overlapped movement patterns between playing positions per pattern mining algorithm were explored.

**Separation of players into playing positions.** The separation of players into playing positions was achieved through machine learning classification modelling. Classification algorithms are usually fitted on a dataset (consisting of independent and dependent variables) to develop a model that can correctly label previously unseen observation(s) into its group (i.e. dependent variable values). This study generated classification input datasets such that the set of unique movement patterns derived from the movement patterns extracted per pattern mining algorithm were the independent variables and each observation represents players per match. The values of each observation are either 1 if players performed the unique movement patterns within fixtures or 0 if otherwise. The value of the dependent variable is either hooker or winger depending on the players' playing position.

Five machine-learning classification algorithms were considered for classification modelling. Decision Tree [27], Gaussian Naive Bayes [28], Random Forest [29], Logistic Regression [30] and Multi-Layered Perceptron (MLP) [31] algorithms were selected because they have different learning methods and fit distinct models. The classification algorithms were implemented in the scikit-learn (version 1.1.3) python module [32]. The parameters for fitting each classification algorithm are presented in Table 3. The classification models were fitted via the $k$-fold cross-validation technique [33]. The cross-validation *n splits* parameter was set to 10, *random state* was set to 10 and the *shuffle* parameter was set to "True". The 10-fold cross-validation technique divides the data into ten chunks in ten iterations and uses nine chunks for training and one separate chunk for testing in each iteration. The cross-validated models' performances were evaluated by aggregated accuracy, precision, recall and f1-score metrics.

Additionally, the feature importance scores [30] of movement patterns used by the best classification models were analysed. This was done to identify top-20 important movement patterns (per pattern mining algorithm) used for classification model development. The source code for computing the Jaccard similarity of unique movement patterns, visualization of overlapped frequent patterns between the pattern mining algorithms and playing positions,

**Table 3. Classification algorithms parameter settings.**

| Algorithms | Parameters |
|---|---|
| Decision Tree | *default* |
| Gaussian Naïve Bayes | *default* |
| Random Forest | random state = 1 |
| Logistic Regression | penalty = "l1", solver = "liblinear" |
| MLP | max iter = 300, random state = 5 |

**Table 4. Jaccard similarity scores of extracted movement patterns per algorithm.**

| Jaccard Similarity Scores | | | |
|---|---|---|---|
| Algorithms | LCCspm | LCS | Apriori Close |
| LCCspm | 1.0 | 0.19 | 0.008 |
| LCS | 0.19 | 1.0 | 0.009 |
| Apriori Close | 0.008 | 0.009 | 1.0 |

machine learning classification model development and evaluation, and feature importance scores analyses are all available publicly and online in a GitHub repository [34].

# Results

## Jaccard analysis

LCCspm algorithm extracted a unique set of 3,881 frequent closed contiguous movement patterns. The LCS algorithm (of the "SMP" framework) extracted a unique set of 2,513 frequent longest common subsequence movement patterns. The AprioriClose algorithm extracted a unique set of 155 frequent closed itemsets movement patterns.

Table 4 reports the results of Jaccard similarity analysis to quantify the similarity in the extracted unique sets of movement patterns between pattern mining algorithms. Overall, Jaccard scores ranged from 0.008 to 0.19 suggesting limited similarity among the movement patterns extracted by the three algorithms.

## Overlap movement patterns

**Overlapping movement patterns between algorithms.**  *LCCspm vs. LCS.* 1022 unique movement pattern overlapped between LCCspm (26% of total) and LCS (40% of total) algorithms. In the most frequent-50 extracted movement patterns for each pattern mining algorithm, 32 movement patterns overlapped between LCCspm and LCS algorithms. Fig 2 highlights the visualisation of the overlapped movement patterns based on the frequency count of LCCspm algorithm between LCCspm and LCS algorithms. The movement patterns "VU" (sprint acceleration backwards and sprint acceleration with large-change of direction),

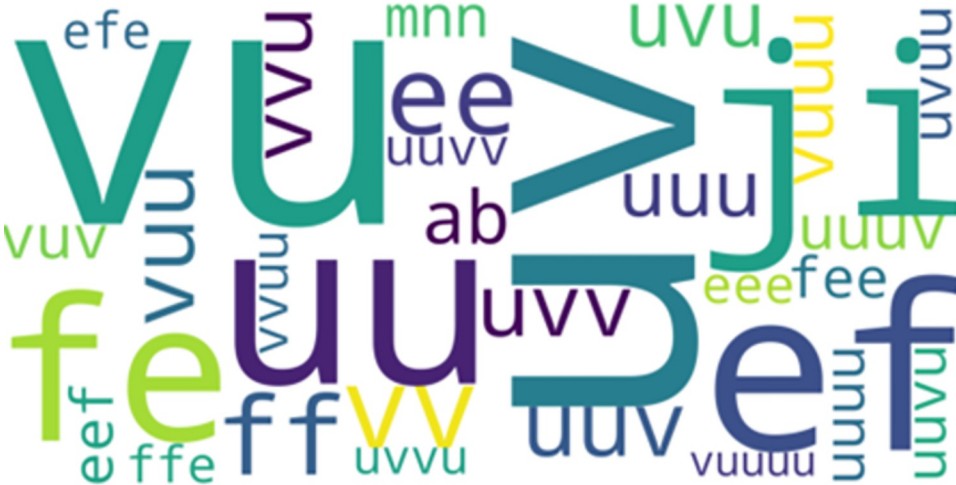

**Fig 2. Overlapped movement patterns between the most frequent-50 LCCspm and LCS patterns.**

"uuv" (jog acceleration straight[x2] and jog acceleration with acute-change of direction), "ji" (walk acceleration with acute-change of direction and walk acceleration straight) and "eef" (walk neutral straight [x2] and walk neutral acute-change of direction) were among the extracted most frequent on-field activities that overlapped between the LCCspm and LCS movement patterns (Fig 2). However, no movement patterns overlapped in the least frequent-50 movement patterns extracted by both LCCspm and LCS algorithms.

*LCCspm vs. AprioriClose.* 32 movement patterns overlapped between the unique sets of movement patterns identified by LCCspm (0.83% of total) and AprioriClose (20.65% of total) algorithms. In the most frequent-50 extracted movement patterns for each pattern mining algorithm, 3 movement patterns overlapped between AprioriClose and LCCspm algorithms. Fig 3 highlights the visualization of the overlapped movement patterns based on the frequency count of AprioriClose algorithm. The movement pattern "uv" (jog acceleration straight and jog acceleration with acute-change of direction is the most frequent followed by "ij" (walk acceleration straight and walk acceleration with acute-change of direction (Fig 3). However, no movement patterns overlapped in the least frequent-50 movement patterns extracted by both AprioriClose and LCCspm algorithms.

*LCS vs. AprioriClose.* 25 movement patterns overlapped between the LCS (1% of total) and AprioriClose (16.13% of total) algorithms. In the most frequent-50 extracted movement patterns for each pattern mining algorithm, 3 movement patterns overlapped between LCS and AprioriClose algorithms. Fig 4 visualises the overlapped movement patterns based on the frequency count of the LCS algorithm. The movement pattern "ef" (walk neutral straight and walk neutral with acute-change of direction) was the second most frequent overlapping pattern (Fig 4). However, no movement patterns overlapped in the least frequent-50 movement patterns extracted by both LCS and AprioriClose algorithms.

**Overlapped frequent-50 movement patterns between positions.** The further analysis of the overlapped movement patterns between the most frequent-50 frequent LCCspm and LCS patterns (Fig 2) by playing positions revealed that hookers performed twenty-nine (29) overlapped patterns (Fig 5) and wingers performed thirty-one (31) overlapped patterns (Fig 6).

The movement patterns "ji" denoted as walk acceleration acute-change and walk acceleration straight and "fee" denoted as walk neutral acute-change and [walk neutral straight] x 2

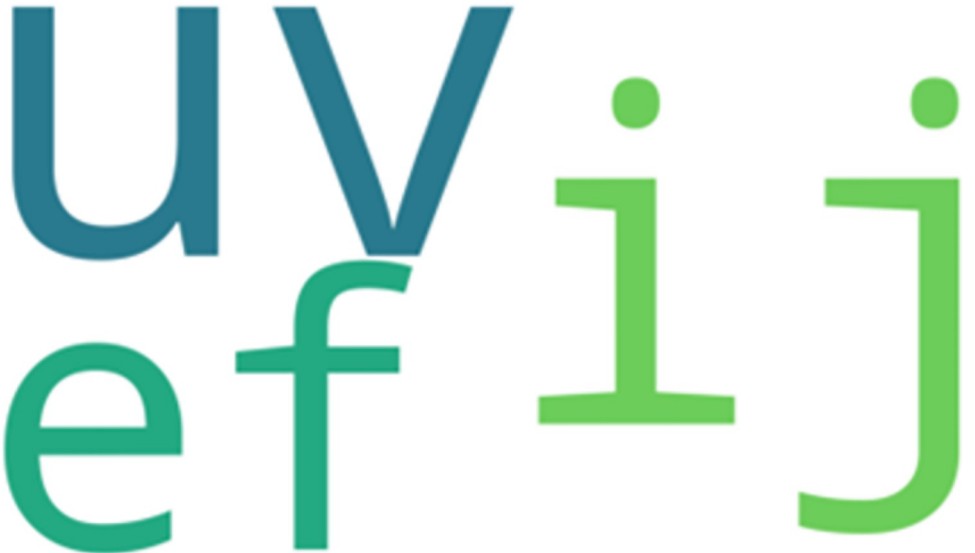

**Fig 3. Overlapped movement patterns between the most frequent-50 AprioriClose and LCCspm patterns.**

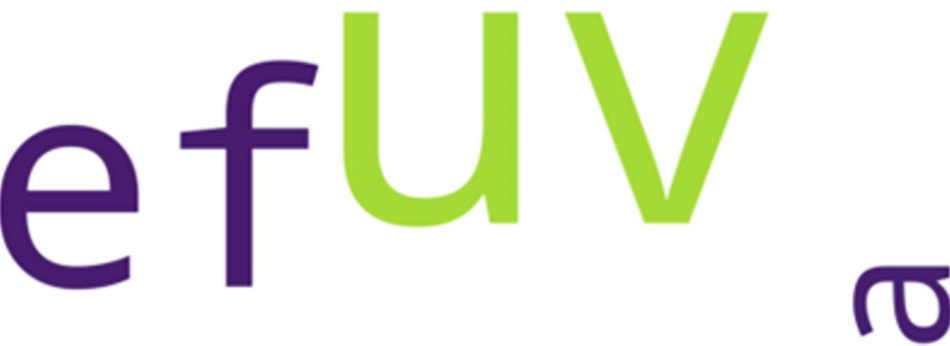

**Fig 4. Overlapped movement patterns between the most frequent-50 LCS and AprioriClose patterns.**

were mainly performed by wingers while movement patterns "uuuuv" denoted as [jog acceleration straight] x 4 and jog acceleration acute-change, and "mn" jog deceleration straight and jog deceleration acute-change were mainly performed by hookers among other overlapped movement patterns.

All overlapped movement patterns ("ef, uv and ij") between the most frequent-50 frequent LCCspm and AprioriClose patterns (Fig 3) were performed by hookers and wingers. Similarly, both hookers and wingers performed the overlapped movement patterns ("ef, uv and a") between the most frequent-50 frequent movement patterns extracted by LCS and AprioriClose algorithms (Fig 4).

**Overlapped movement patterns between positions per algorithm.** *LCCspm.* 2,282 and 3,174 sets of frequent closed contiguous movement patterns were identified by LCCspm to profile hookers and wingers respectively. A total of 1,575 movement patterns overlapped between both playing positions (visualized in Fig 7 based on how often they were performed by hookers and Fig 8 based on how often they performed by wingers) as extracted by LCCspm algorithm.

Also, LCCspm profiled 707 closed contiguous movement patterns uniquely performed by hookers and another set of 1599 movement patterns performed only by wingers.

*LCS.* 1,534 and 1,632 sets of longest common movement patterns were identified by the LCS algorithm of the "SMP" framework to profile hookers and wingers respectively. A total of

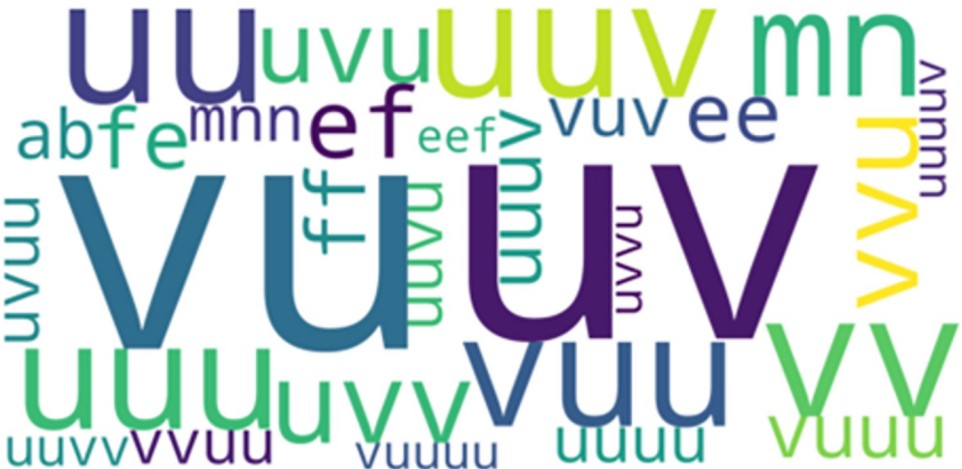

**Fig 5. LCCspm and LCS overlap movement patterns performed by hookers.**

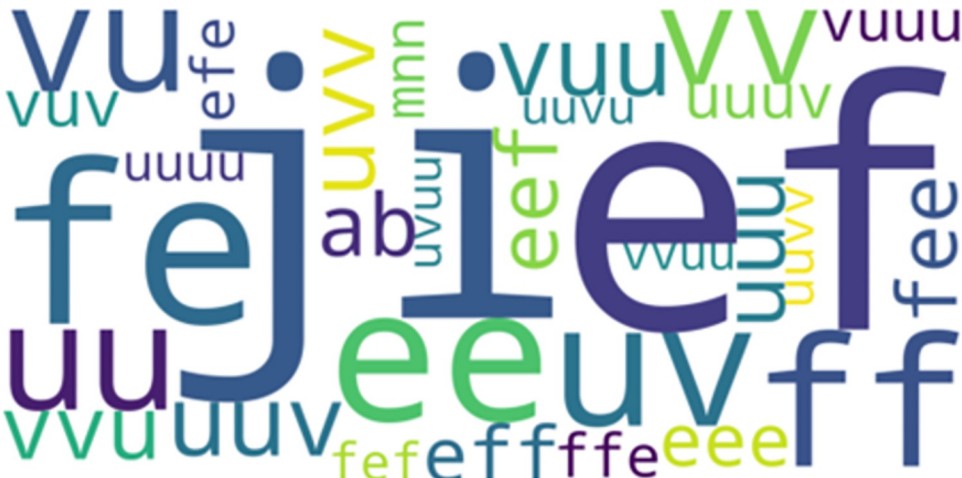

**Fig 6. LCCspm and LCS overlap movement patterns performed by wingers.**

653 overlapped movement patterns were identified between hookers and wingers (visualized in Fig 9 based on often they were performed by hookers and Fig 10 based on how often they were performed by wingers) as extracted by the LCS algorithm.

The LCS algorithm of the "SMP" framework profiled 818 longest common movement patterns performed only by hookers and another set of 979 movement patterns performed only by wingers.

*AprioriClose*. 142 and 136 sets of non-sequential movement patterns were identified by the AprioriClose algorithm to profile hookers and wingers respectively. A total of 123 overlapped movement patterns were identified between both hookers and wingers (visualized in Fig 11 based on often they were performed by hookers and Fig 12 based on how often they were performed by wingers) as extracted by the AprioriClose algorithm.

AprioriClose algorithm profiled a total of 19 non-sequential movement patterns performed only by hookers and another set of 13 non-sequential movement patterns performed only by wingers.

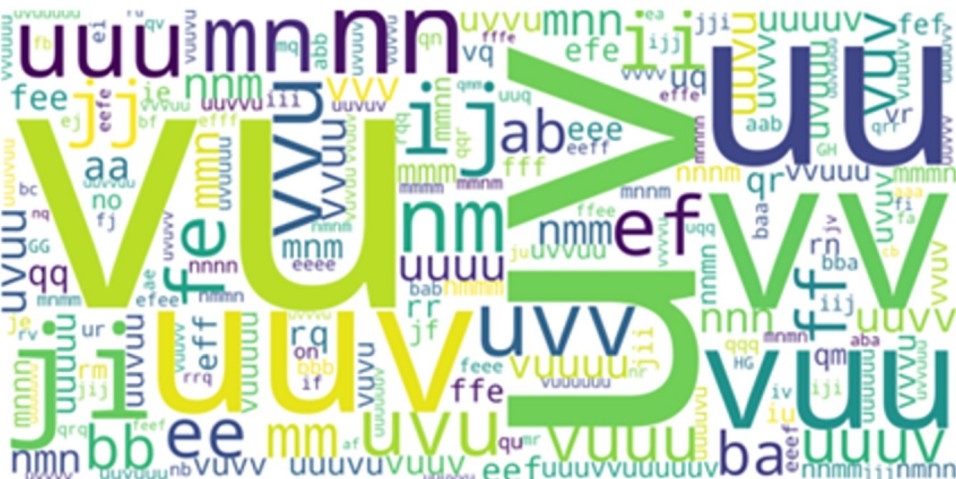

**Fig 7. LCCspm overlapped movement patterns as performed by hookers.**

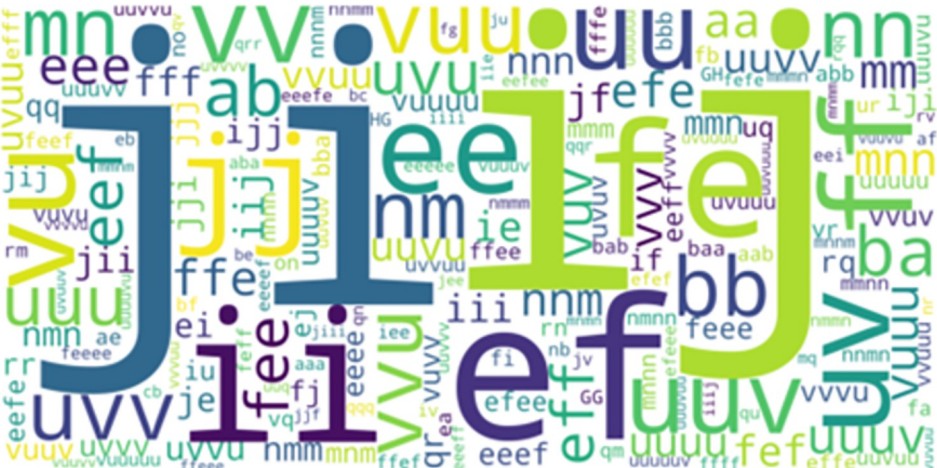

**Fig 8. LCCspm overlapped movement patterns as performed by wingers.**

## Separation of players into playing positions

Three datasets were generated for classification modelling. The first dataset (representing LCCspm algorithm) contained 3,881 independent variables. The second dataset (representing LCS algorithm) contained 2,849 independent variables. The third dataset (representing AprioriClose algorithm) contained 155 independent variables.

The accuracy of the selected five (5) machine learning classification algorithms after modelling on all three datasets are reported in Table 5. All classifiers fitted on the LCCspm dataset achieved the highest accuracies when compared to their counterparts fitted on the LCS and AprioriClose datasets. For example, the Decision Tree classifier achieved an accuracy of 82.83% on the LCCspm dataset compared to 56.36% accuracy on the LCS dataset and 73.56% accuracy on the AprioriClose dataset.

The MLP classifier fitted on the dataset having LCCspm movement patterns as its independent variables had the highest individual accuracy of 91.02% among all other classifiers fitted

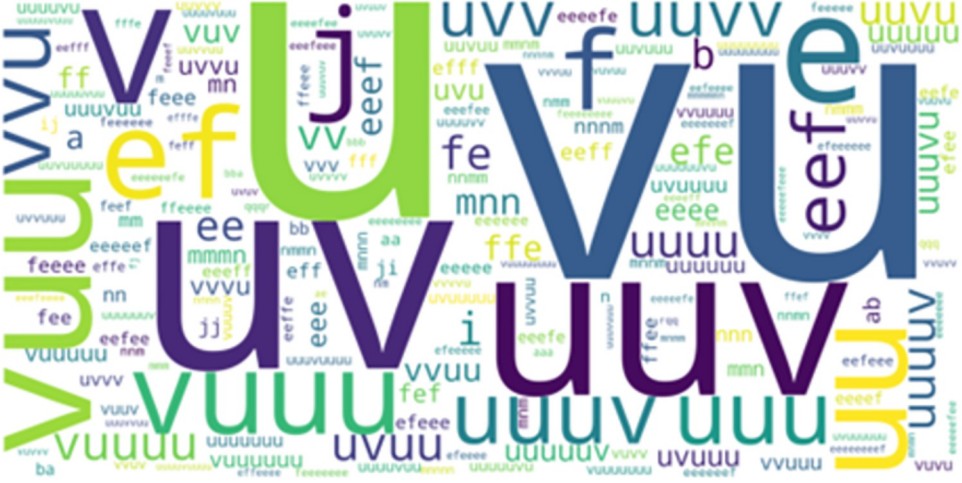

**Fig 9. LCS overlapped movement patterns as performed by hookers.**

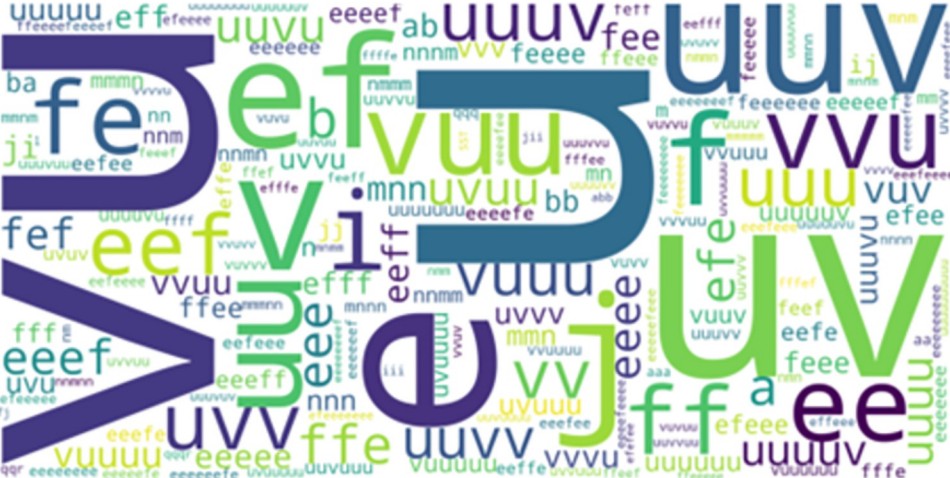

**Fig 10. LCS overlapped movement patterns as performed by wingers.**

on any of the three datasets. MLP classifier achieved 61.78% and 80.9% accuracies on the LCS and AprioriClose datasets respectively. Meanwhile, the accuracy of the Gaussian Naive Bayes classifiers is the lowest among other classification algorithms, across all algorithms.

Consequently, the LCCspm algorithm used for mining closed contiguous movement patterns provided the most data-driven insights for separating players into playing positions based on the classification models' performances. The AprioriClose algorithm used for mining closed itemsets movement patterns provided the second-best data-driven insights (among three selected pattern mining algorithms) to separate players into playing positions. Meanwhile, the LCS algorithm of the "SMP" framework ranked provided the least data-driven insights to separate players into playing positions.

From Table 5, the Logistic Regression algorithm fitted two of the three most accurate classification models per pattern mining algorithm. It fitted the most accurate classification models on the AprioriClose and LCS datasets, the accuracy of 82.95% and 65.83% respectively. Meanwhile, it fitted the second-best accurate classification model of 89.77% accuracy on the

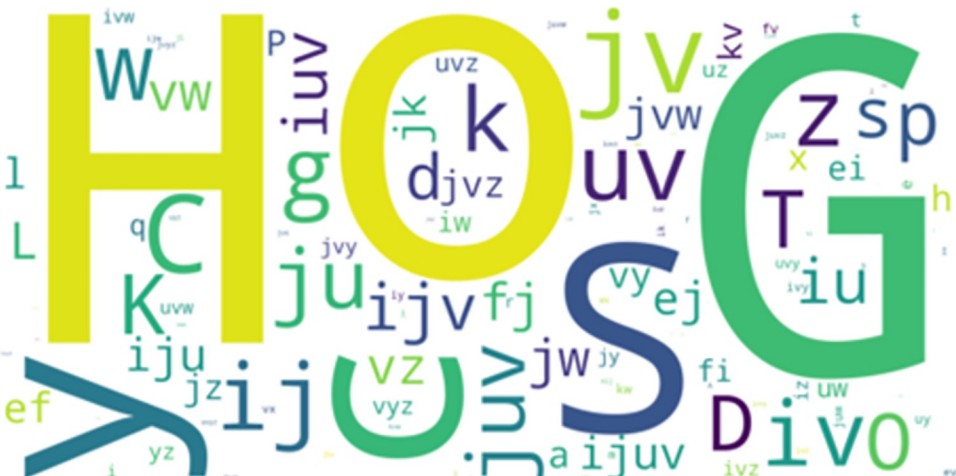

**Fig 11. AprioriClose overlapped movement patterns as performed by hookers.**

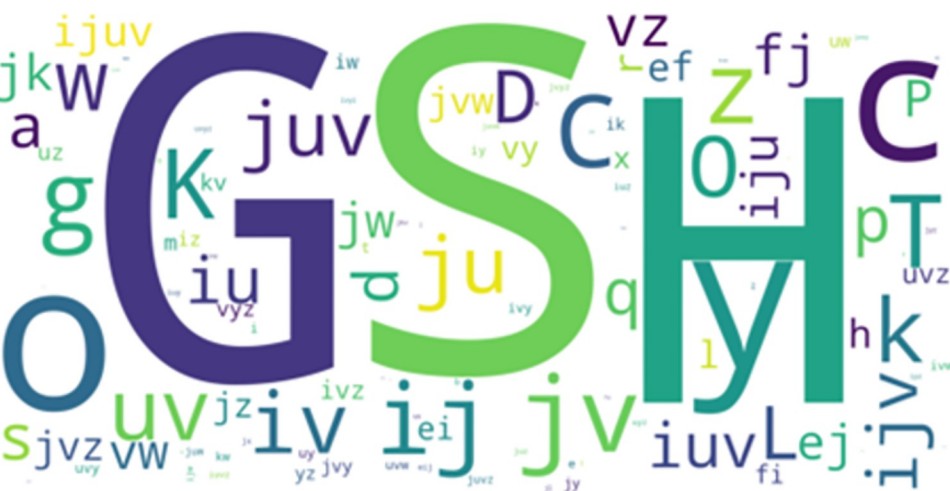

**Fig 12. AprioriClose overlapped movement patterns as performed by wingers.**

LCCspm dataset. As such, further analysis for the top-20 feature importance scores of the movement patterns used by the Logistic regression models per pattern mining algorithm was conducted and reported in Table 6.

## Discussion

This study is the first to identify which pattern mining algorithms (LCCspm, LCS, Apriori-Close) provide the best capable set of movement patterns to classify rugby league hookers and wingers' playing positions. A secondary aim was to understand the similarity of extracted movement patterns among all three algorithms and between the two playing positions (hookers and wingers). Hookers' and wingers' playing positions were chosen as the criterion positions to compare algorithms given their unique tactical and physical roles in professional rugby league. Overall, the findings suggest that the LCCspm pattern mining algorithm

**Table 5. Classifiers' separation accuracies using sets of extracted movement patterns.**

| Classifier | Accuracy (%) | Precision | Recall | F1 Score | Algorithm |
|---|---|---|---|---|---|
| Decision Tree | 82.83 | 0.83 | 0.83 | 0.83 | **LCCspm** |
| Gaussian Naïve Bayes | 58.09 | 0.73 | 0.58 | 0.5 | |
| RandomForest | 88.61 | 0.89 | 0.89 | 0.89 | |
| Logistic Regression | 89.77 | 0.9 | 0.9 | 0.9 | |
| MLP | **91.02** | 0.91 | 0.91 | 0.91 | |
| Decision Tree | 57.72 | 0.58 | 0.58 | 0.57 | LCS |
| Gaussian Naïve Bayes | 50 | 0.51 | 0.51 | 0.47 | |
| RandomForest | 63.8 | 0.64 | 0.64 | 0.64 | |
| Logistic Regression | 65.83 | 0.66 | 0.66 | 0.66 | |
| MLP | 61.78 | 0.62 | 0.62 | 0.62 | |
| Decision Tree | 73.56 | 0.74 | 0.74 | 0.73 | AprioriClose |
| Gaussian Naïve Bayes | 54.36 | 0.6 | 0.53 | 0.43 | |
| RandomForest | 81.76 | 0.82 | 0.82 | 0.82 | |
| Logistic Regression | 82.05 | 0.82 | 0.82 | 0.82 | |
| MLP | 80.90 | 0.81 | 0.81 | 0.81 | |

**Table 6. Logistic Regression Top 20 important patterns and scores per algorithm.** (a) Top 20 APR Patterns Importance Score. (b) Top 20 SMP Patterns Importance Score. (c) Top 20 LCC Patterns Importance Score.

(a)

| APR Patterns | Importance Score |
|---|---|
| q | 2.12 |
| G | 2.03 |
| ik | 1.59 |
| P | 1.44 |
| juv | 1.24 |
| h | 1.10 |
| L | 1.10 |
| jvz | 0.99 |
| iuvy | 0.90 |
| o | 0.85 |
| iju | 0.85 |
| ijw | 0.82 |
| ijvz | 0.81 |
| jk | 0.76 |
| iuz | 0.69 |
| uvz | 0.55 |
| juw | 0.49 |
| juvw | 0.49 |
| ivz | 0.48 |
| jl | 0.45 |

(b)

| SMP Patterns | Importance Score |
|---|---|
| eeeeefeeeeeeee | 1.95 |
| ii | 1.57 |
| ff | 1.41 |
| eee | 1.40 |
| ee | 1.38 |
| fee | 1.28 |
| uuvvu | 1.26 |
| qqqqqq | 1.15 |
| vuvvu | 1.11 |
| eeeeeeefee | 1.10 |
| fe | 1.07 |
| nnmn | 1.00 |
| ef | 0.95 |
| j | 0.94 |
| mmmnn | 0.89 |
| GG | 0.84 |
| eeeee | 0.77 |
| i | 0.76 |
| uuvvvu | 0.76 |
| efeeeeeee | 0.76 |

(c)

| LCC Patterns | Importance Score |
|---|---|
| iiiii | 2.02 |

(*Continued*)

**Table 6.** (Continued)

| | |
|---|---|
| SS | 1.67 |
| iiie | 1.40 |
| GGG | 1.28 |
| qrq | 1.17 |
| iji | 1.16 |
| jjji | 1.13 |
| ijjj | 0.95 |
| HH | 0.93 |
| mmr | 0.91 |
| iii | 0.85 |
| uvvuuv | 0.83 |
| vuvuu | 0.83 |
| zz | 0.82 |
| jie | 0.81 |
| GGSS | 0.79 |
| iie | 0.78 |
| jjf | 0.77 |
| fe | 0.73 |
| ijii | 0.72 |

provided the best set of movement patterns for separating hookers and wingers and that there is a lack of similarity in the extracted movement patterns between algorithms.

The classification results of this study revealed the extent of separating players into playing positions, based on each set of frequent movement patterns, extracted from the same sets of movement sequences, and under the same parameter condition. Table 5 shows that the separation of elite rugby players into playing positions (i.e., hookers and wingers) based on their frequent movement patterns is best done using their extracted closed contiguous movement patterns, profiled by LCCspm algorithm. The LCCspm closed contiguous movement pattern using the Multi-Layered Perceptron classifier performed best to classify hookers and wingers in professional rugby league, with an overall accuracy of 91.02%. The AprioriClose closed itemsets (non-consecutive) movement patterns offered a better separation accuracy than the longest common subsequence movement patterns of the LCS algorithm. AprioriClose movement patterns provided a decent separation (through Logistic Regression accuracy of 82.05%). Its lowered accuracy can be attributed to the nature of its movement patterns being non-consecutive, non-sequential and without repeated movement activity. Also, the results of this study indicate player movement profiling using the LCCspm algorithm will discover more numbers of movement patterns for profiling players from the same sets of movement sequences than AprioriClose and LCS algorithms. This implies there are more discoverable consecutive movement patterns than non-consecutive and non-sequential movement patterns. More so, Jaccard similarity scores (1 being full similarity) ranged from 0.008 to 0.19 among movement patterns algorithms (Table 4), suggesting a lack of similarity in the extracted patterns overall. The lack of similarity among the sets of movement patterns can be attributed to the pattern mining algorithms as they extract consecutive, non-consecutive, and non-sequential movement patterns respectively. LCCspm and LCS sets of movement patterns shared a relatively higher similarity because both algorithms extract some form of sequential movement patterns as opposed to AprioriClose non-sequential patterns. Based on these results, the

LCCspm algorithm is justified and identified as the best for profiling movement patterns of rugby league players into hookers and wingers playing positions.

The application of LCCspm algorithm to profile the movement of hookers and wingers revealed wingers performed 892 movement patterns more than hookers. This suggests a more variable movement profile of wingers than hookers. There were overlapped movement patterns between hookers and wingers (Fig 6), but the frequency at which the movement patterns were performed differs by playing positions.

Overlapped movement patterns with a combination of movement units *u* and *v* which indicates accelerated jogs with some acute change in direction or on straights were mostly performed by hookers (Fig 7). Wingers on the other hand performed overlapped movement patterns that included accelerated walks with some acute direction changes as indicated by movement units *j* and *i* (Fig 8).

This study also identified groups of movement activities performed uniquely by hookers and wingers. For example, the LCCspm algorithm identified hookers as the only positional group that performed the sequential movement pattern "GGGGGGGGGGGGGSSSSSSS" (Run-Acceleration-Straight [x12] and Sprint-Acceleration-Straight [x7]). Equally, only wingers completed the sequential movement pattern of "TSSTSTTSST" (Sprint-Acceleration-Acute change, Sprint-Acceleration-Straight [x2], Sprint-Acceleration- Acute change, Sprint-Acceleration-Straight, Sprint-Acceleration-Acute change [x2], Sprint-Acceleration-Straight [x2], Sprint-Acceleration-Acute change). It is well established that wingers complete greater high-speed ($>5m.s^{-1}$) activity during matches than hookers (wingers: 626m vs. hookers: 285m) [35], although these differences are less pronounced with global acceleration-based measures (e.g., average acceleration over a period of time). These differences are likely due to the vastly different tactical roles of wingers (e.g., returning kicks in attack leading to open space to move at high speed) vs. hookers (e.g., repositioning behind the play the ball to distribute possession). Applying pattern mining algorithms to uncover the sequential nature of the occurrences of activity enables the better capability to classify positional groups and aid in enhanced training specificity.

It is also noteworthy that the most important variables used by the Logistic Regression classification model are mostly not part of the most frequent-50 overlapping patterns profiled by the LCCspm algorithm. This indicates that the not-too-frequent movement patterns and those uniquely performed by players of each playing position provided insights used for players' playing position separation. The twenty most important LCCspm movement patterns used for fitting the Logistic Regression classifier consist of 2 to 6-length on-field movement activities (Table 6C). The second most important variable "SS" (denoted as [sprint acceleration straight] x2) and ninth most important variable "HH" (denoted as [run acceleration acute-change] x2) discovered by the LCCspm pattern mining algorithm are the only patterns to include on-field activities "S" and "H" in its set of most important movement patterns across all three pattern mining algorithms. The nineteenth important movement pattern "fe" in Table 6C is the only pattern present in the most frequent LCCspm and LCS overlapped movement pattern in Fig 2). Given that the movement patterns extracted by the LCCspm algorithm achieved the highest accuracy for player position classification compared to other patterns, it is concluded that LCCspm and closed contiguous movement patterns are optimal for profiling rugby league players. In practice, sports performance analysts and or data scientists are encouraged to extract players' closed contiguous movement patterns when conducting player profiling analysis. The consecutive sequence of performed activities by players are vital for profiling and distinguishing players of different playing positions. LCCspm closed contiguous patterns effectively and efficiently captures the consecutiveness of players' movement.

## Conclusions and future works

This study is the first to identify the best pattern mining algorithm and its set of movement patterns for player movement profiling. Closed contiguous movement patterns are the best to separate rugby league hookers and wingers into playing positions because all classification models were most accurate on the dataset generated with LCCspm unique movement patterns as independent variables. Therefore, mining closed contiguous movement patterns for profiling the on-field activities of players is recommended. LCCspm and LCS algorithms extracted movement patterns that shared some form of similarity while AprioriClose movement patterns shared no similarity because itemset does not consider the order of item appearance. Given that one of the cores of sports analytics is the ability to predict [36] sports outcomes or groups, the LCCspm algorithm for mining movement patterns for player position classification is recommended as a useful advanced analytics (predictive) tool for sports analytics. Additionally, this study's method can be replicated for other use cases, such as the extraction of match event patterns. In the future, the identification of the minimum number of patterns to ably separate between groups will be considered. Future consideration will also be given to using the patterns extracted from SMP framework condensed sequences and those extracted through the "LCCspm" algorithm to understand the locomotive and match demand on players and teams.

## Acknowledgments

The authors would like to acknowledge The Rugby Football League (RFL) for the access to GPS data.

## Author Contributions

**Conceptualization:** Victor Elijah Adeyemo, Anna Palczewska, Ben Jones, Dan Weaving.

**Data curation:** Victor Elijah Adeyemo.

**Investigation:** Victor Elijah Adeyemo.

**Methodology:** Victor Elijah Adeyemo.

**Project administration:** Victor Elijah Adeyemo.

**Supervision:** Anna Palczewska, Ben Jones, Dan Weaving.

**Visualization:** Victor Elijah Adeyemo.

**Writing – original draft:** Victor Elijah Adeyemo.

**Writing – review & editing:** Anna Palczewska, Ben Jones, Dan Weaving.

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
