## [Decision Letter · Decision Letter 0]

23 Jan 2024

PONE-D-23-05367Identification of pattern mining algorithm for rugby league players positional groups separation based on movement patternsPLOS ONE

Dear Dr. Adeyemo,

Thank you for submitting your manuscript to PLOS ONE. After careful consideration, we feel that it has merit but does not fully meet PLOS ONE’s publication criteria as it currently stands. Therefore, we invite you to submit a revised version of the manuscript that addresses the points raised during the review process.

We look forward to receiving your revised manuscript.

Kind regards,

Ersan Arslan, Ph.D.

Academic Editor

PLOS ONE

Journal Requirements:

   "No"

Reviewers' comments:

Reviewer's Responses to Questions

**Comments to the Author**

1. Is the manuscript technically sound, and do the data support the conclusions?

Reviewer #1: Yes

Reviewer #2: Yes

Reviewer #3: Yes

2. Has the statistical analysis been performed appropriately and rigorously? 

Reviewer #1: Yes

Reviewer #2: Yes

Reviewer #3: I Don't Know

3. Have the authors made all data underlying the findings in their manuscript fully available?

Reviewer #1: No

Reviewer #2: Yes

Reviewer #3: Yes

4. Is the manuscript presented in an intelligible fashion and written in standard English?

Reviewer #1: Yes

Reviewer #2: Yes

Reviewer #3: Yes

5. Review Comments to the Author

Reviewer #1: This article applies three different pattern mining algorithms to private rugby league data, using the output of these algorithms with five different machine learning classification algorithms to determine whether the different movement patterns could be used to distinguish between hookers and wingers. A more complete description of the data used would be welcome (e.g., how many teams were represented? How much playing time did the players have? How was data cleaned and were there any exclusion criteria?), but the described experiments seem sound and the conclusions could be beneficial in other related areas.

Some more minor recommendations include:

* In the "Data and Processing" subsection of the Method, Figure 1 is referenced, when it should be Table 1.

* In "Separation of Players into Playing Position", "set unique of movement patterns" should be "set of unique movement patterns".

* Table 3 mentions that "default" parameter values were used for some of the tested algorithms. Please move the detail that scikit-learn was used to the sentence that references this table so it is understood that it is the defaults from this library that are used.

* In "Overlapped movement patterns between positions per algorithm", "based on often" should be "based on how often" (a few instances of this)

* Table 1 describes 10 unit movement characters, but several more appear to be used later in the paper. A description of all unit movement characters should be included.

* A figure is referenced as Fig ?? in the "Discussion"

* "noteworthy to point out" seems redundant - replace with "noteworthy"

Reviewer #2: GENERAL COMMENTS

The authors have presented an analytics-focused study examining the performance and similarity of identifying movement patterns in rugby. The methods are well justified based on previous pattern classification works, and the idea of comparing algorithms is sound. Where I see the major weaknesses of the paper in present form is in the results and discussion, namely: (1) the readability and interpretability of many of the results presented is not easy or intuitive (see specific comments for this section below); and (2) the practical implications of what was found (e.g. lack of similarity in algorithms) is potentially not discussed in enough detail. I think once addressed this paper can progress understanding in this area and provide some useful recommendations for future pattern mining and classification work in sport.

I recently received a review on one of my own papers where the reviewer provided an ‘about me’ section, which I found quite valuable in placing additional context around their comments, so it is something that I am trying to add to my own reviews:

I have a research background primarily in sports injury and performance biomechanics, but have a secondary research focus on using data and analytics approaches to characterise and improve sports performance. My work in this area has predominantly focused on netball, with a little in Australian football. I believe I have a basic understanding of the algorithms and approaches used in this paper – but wouldn’t call myself an expert in all of these.

I am also open to being identified on papers I reviewed – so am signing off my review here.

Aaron Fox

Centre for Sport Research, Deakin University

Please see below for specific comments. As no line numbers are present on the submission I’ve tried to use the page numbers and rough section on the page, or used a quote to indicate where the comment relates to.

SPECIFIC COMMENTS

Abstract

In the abstract it is a little unclear how the “best set of movement patterns” is identified based on the three algorithms. I’m expecting more in the main methods, but some indication in the abstract on how this is evaluated would be useful.

Similarly, it is stated that “AprioriClose…shared no significant similarity” with the other algorithms – but it’s unclear how this was determined. Including the Jaccard score here between these algorithms would provide a contrast to the other value presented in the prior sentence. I think this is particularly relevant to establish, as the Jaccard score for the previous comparison (i.e. 0.19) isn’t exactly high for this metric.

Prior to mentioning the results from the Multi-layered perceptron algorithm, this and it’s associated metrics (i.e. precision, recall, F1 score) are not mentioned in the abstract – so it sort of comes out of the blue. Presumably these relate to the classification algorithms mentioned earlier in the abstract – but some additional clarity around these methods, so that the subsequent results make sense in the abstract would be useful.

Introduction

The introduction provides a good summary of the area and justification for the study. I thought that there could be more of an emphasis on the practical justification of the study (i.e. why movement patterns are important) over the detailed analysis of the pattern mining algorithms – but can potentially see the need for this given the study is more so focused on algorithms over practical implications in rugby at this point. I would still consider shifting the emphasis slightly – but would leave this up to the authors.

Page 2/16: “However, the insights are currently based on aggregated physical, technical and tactical demands either across a whole match or for specific periods within the game.” A follow-up sentence to this explaining why this is an issue would help set-up the subsequent notion of investigating pattern mining algorithms. This is somewhat touched on later (on page 3/16), but I feel that this statement would work better up here.

Page 2/16: “Nowadays, sequential pattern mining algorithms are applied to sports data to profile players’ movements.” – some citation to relevant literature to support this would be nice.

Page 2-3/16: “Also, the algorithm was reportedly developed because the existing Closed Contiguous Sequential Pattern mining (CCSpan) algorithm may not produce usable movement patterns for sporting contexts and could not scale well on large sets of (lengthy) players’ movement sequences.” – I’m a big fan of active writing and taking ownership over the previous work a group of authors has conducted. Given this is past paper you’re referring to is your work, I’d recommend referring to it in this way (i.e. rather than “was reportedly developed…” – consider “we developed…”).

Page 3/16: “However, the investigation of which type of movement pattern is best for rugby league players’ movement profiling is yet to be explored” – should this read “which type of movement pattern mining algorithm is best…”?

Methods

Given there are numerous aims and methods associated with these, I think the back-end of the introduction could use some more solid statements around what these specific aims are (e.g. “The specific aims of this study are (1) …; (2) …”). This could then link into the ‘overview’ section of the methods by linking up these aims to the various approaches described (i.e. “For aim 1…we did this”; for aim 2…” etc). I feel this would help with clarity around what the various algorithms and metrics are being used for.

Page 4/16: “A total of 1,036 total observations were included which represent players’ movement sequence per fixture.” Some details (perhaps as supplementary info) around how the players data is spread across these observations could provide additional context on the dataset (e.g. presumably the 22 hookers didn’t play every game? Or maybe they did?).

Page 5/16: “The studies [12, 17] used similar parameter values to enable the extraction of large and

longer-length frequent patterns.” It would help to clarify the sports these are in – it looks like they are both rugby? Which would provide further support for replicating these methods.

Page 5/16: “a set of unique movement patterns was derived by computing the union of all sets of extracted movement patterns.” I’m a little unclear of what this refers to/how it was done. Some extra detail on these methods would help others replicate these analyses.

Page 6/16: The section label (and subsequent use of this terminology) ‘Pattern Mining Algorithms (Movement Patterns) Validation’ – with particular reference to the use of ‘validation’ – seems slightly incorrect to me. To validate these pattern mining algorithms, you would need to know what the ‘real’ patterns are, whereas it seems the paper assumes that the algorithms are simply extracting the true or real patterns. ‘Agreement’ seems like a better term over ‘validation.’

Page 6/16: “For each pair, the most frequent-50 and least frequent-50 extracted movement patterns of each pattern mining algorithm were checked for overlapping and visualised.” Why the most and least frequent 50? Why not more or less? This seems a little subjective and could use some further justification.

Page 7/16: “The cross-validated models’ performances were evaluated by aggregated

accuracy, precision, recall and f1-score metrics” – it’s unclear if this was done on a separate ‘test’ dataset outside of the training set. If it wasn’t, then this needs to be clear – and perhaps it’s worth considering doing so to truly validate these models on ‘unseen’ data.

Results

Currently the main issue I see with the results is the difficulty in following some ideas due to the lack of intuitiveness in interpreting the results. When I saw the word-clouds as figures in the PDF document, I initially thought they were a watermark-like placeholder and I needed to download the figure – but once I had downloaded the figure I realised what they were. These word-cloud figures aren’t easily interpretable (particularly given that out of context, the text strings don’t really mean anything) – so I think a different approach to presenting these would be appropriate. If keeping some sort of visual representation of movement patterns is desired (see my next comment on this though), it would be nice if you could make these visuals more intuitive and combine a lot of this information into one figure/table (i.e. combining Fig. 2-6 to show overlap across the three algorithms and when split by position). I don’t think this is an easy task, and may even be unachievable (but see my next comment for why I don’t think this matters).

Similar to above, I don’t necessarily think the entire sections on comparing more individualised movement patterns between algorithms adds that much. The idea behind metrics like the Jaccard index is that it provides this summary-style quantitative metric to demonstrate similarity – which is easy to understand. Probing through individual pattern similarities like is done across most of the results section is quite difficult to follow and interpret meaning from. If you wish to probe these more individualised comparisons, I strongly suggest finding a more summary-like way of doing this to make these comparisons more digestible for readers. I think this idea is demonstrated in your summary paragraph at the beginning of the discussion – where you state “Overall, the findings suggest that the LCCspm pattern mining algorithm provided the best set of movement patterns for separating hookers and wingers and that there is a lack of similarity in the extracted movement patterns between algorithms” – which I’d argue has been predominantly determined based on the Jaccard score and classification models (i.e. these main findings haven’t really been determined based on individual movement pattern comparisons).

Similar to the figures, I’m not sure how much added context Table 6 adds given that the pattern indicators aren’t intuitive to readers – i.e. “eeeeefeeeeeeeee” could be the noise someone makes when dropping something heavy on their toe. Based on this I think Table 6 could be removed.

Discussion

Page 12/16: “More so, Jaccard similarity scores (1 being full similarity) ranged from 0.008 to 0.19 among movement patterns algorithms (Table 4), suggesting a lack of similarity in the extracted patterns overall. LCCspm and LCS sets of movement patterns shared higher similarity because both algorithms extract some form of sequential movement patterns as opposed to AprioriClose non-sequential patterns. Based on these results, the LCCspm algorithm is justified and validated as the best for profiling movement patterns of rugby league players into hookers and wingers playing positions.” I think further discussion on the lack of similarity between algorithms is worthwhile. Even the Jaccard index of 0.19 is relatively low, and this probably has implications in research and practical applications where different algorithms are used. For example, two studies that use different algorithms could come to completely different conclusions on what the movement patterns in players of a given sport are. Some further explanation of why these differences happen, and discussion of these implications would be relevant. The importance of this links into a later statement on page 13/16 – “Applying pattern mining algorithms to uncover the sequential

nature of the occurrences of activity enables the better capability to classify positional

groups and aid in enhanced training specificity.” With different algorithm use, these identified patterns will likely differ, which might impact on this training specificity you refer to.

Page 13/16: “This further validates LCCspm algorithm and closed contiguous movement patterns as the best pattern to profile rugby league players into playing positions.” I’m not sure the preceding information here links to this statement – i.e. I don’t think the features included as more important in these models supports one as being better over the other, only the classification accuracy can really be used to support this.

Reviewer #3: Thank you for the opportunity to review this well written manuscript.

I have only minor revisions/questions to be answered before I believe this manuscript is acceptable for publication.

1. Could different cohorts e.g., southern Hempishere or women's players be incorporated?

2. How can the results of your study be used practically? e.g. can you add more detail into the discussion regarding how practitioners can apply your findings?

6. PLOS authors have the option to publish the peer review history of their article (what does this mean?). If published, this will include your full peer review and any attached files.

Reviewer #1: No

Reviewer #2: **Yes: **Aaron Fox

Reviewer #3: **Yes: **Sean Scantlebury

---

## [Author Response · Author response to Decision Letter 0]

13 Feb 2024

Response to reviewers' comments are provided in the uploaded "Response to Reviewers.docx" file.

---

## [Decision Letter · Decision Letter 1]

20 Mar 2024

Identification of pattern mining algorithm for rugby league players positional groups separation based on movement patterns

PONE-D-23-05367R1

Dear Dr. Adeyemo,

We’re pleased to inform you that your manuscript has been judged scientifically suitable for publication and will be formally accepted for publication once it meets all outstanding technical requirements.

An invoice for payment will follow shortly after the formal acceptance. To ensure an efficient process, please log into Editorial Manager at Editorial Manager® , click the 'Update My Information' link at the top of the page, and double check that your user information is up-to-date. If you have any billing related questions, please contact our Author Billing department directly at authorbilling@plos.org.

Kind regards,

Ersan Arslan, Ph.D.

Academic Editor

PLOS ONE

Additional Editor Comments (optional):

Reviewers' comments:

Reviewer's Responses to Questions

**Comments to the Author**

1. If the authors have adequately addressed your comments raised in a previous round of review and you feel that this manuscript is now acceptable for publication, you may indicate that here to bypass the “Comments to the Author” section, enter your conflict of interest statement in the “Confidential to Editor” section, and submit your "Accept" recommendation.

Reviewer #1: All comments have been addressed

Reviewer #2: All comments have been addressed

2. Is the manuscript technically sound, and do the data support the conclusions?

Reviewer #1: Yes

Reviewer #2: Yes

3. Has the statistical analysis been performed appropriately and rigorously? 

Reviewer #1: Yes

Reviewer #2: Yes

4. Have the authors made all data underlying the findings in their manuscript fully available?

Reviewer #1: No

Reviewer #2: Yes

5. Is the manuscript presented in an intelligible fashion and written in standard English?

Reviewer #1: Yes

Reviewer #2: Yes

6. Review Comments to the Author

Reviewer #1: The authors have addressed my comments sufficently.

One minor suggestion from the new edits is to replace "and they been widely used" with "and they have been widely used"

Reviewer #2: Thank you for addressing all comments provided by myself and other reviewers. These have been answered in an appropriate fashion and hence I would be happy for the paper to be accepted in it's current format.

My one remaining suggestion still revolves around the majority of figures and tables that have remained in the manuscript. While you have made a relevant argument for why these are there, as an external reader I still find them quite unintuitive and difficult to comprehend the meaning behind them. Given how info heavy this paper is, my feeling would be to minimise any unintuitive information from the main paper (e.g. have these figures as supplementary material). This is not 'do this to get my acceptance' type demand - just a suggestion from an interested external reader.

Aaron Fox

Centre for Sport Research

Deakin University

7. PLOS authors have the option to publish the peer review history of their article (what does this mean?). If published, this will include your full peer review and any attached files.

Reviewer #1: No

Reviewer #2: **Yes: **Aaron S Fox

---

## [Editor Report · Acceptance letter]

29 Mar 2024

PONE-D-23-05367R1 

PLOS ONE

Dear Dr. Adeyemo, 

I'm pleased to inform you that your manuscript has been deemed suitable for publication in PLOS ONE. Congratulations! Your manuscript is now being handed over to our production team.

Kind regards, 

on behalf of

Dr. Ersan Arslan 

Academic Editor

PLOS ONE